# Corrosion Behavior of Chromium Coated Zy-4 Cladding under CANDU Primary Circuit Conditions

Diana Diniasi [1,2], Florentina Golgovici [2,*], Alexandru Anghel [3], Manuela Fulger [1], Carmen Cristina Surdu-Bob [3] and Ioana Demetrescu [2,4,*]

1  Institute for Nuclear Research Pitesti, POB 78, Campului Street, No. 1, 115400 Mioveni, Romania; diana.diniasi@nuclear.ro (D.D.); manuela.fulger@nuclear.ro (M.F.)
2  Department of General Chemistry, University Politechnica of Bucharest, Splaiul Independentei Street, No. 313, 060042 Bucharest, Romania
3  National Institute for Laser, Plasma and Radiation Physics, Atomistilor Street, 077126 Magurele, Romania; alexandru.anghel@inflpr.ro (A.A.); cristina.surdubob@plasmacoatings.ro (C.C.S.-B.)
4  Academy of Romanian Scientists, 3 Ilfov, 050094 Bucharest, Romania
*  Correspondence: florentina.golgovici@upb.ro (F.G.); i_demetrescu@chim.upb.ro (I.D.); Tel.: +40-214-023-930 (F.G.)

**Abstract:** The manuscript is focused on corrosion behavior of a Cr coating under CANada Deuterium Uranium(CANDU) primary circuit conditions. The Cr coating is obtained via the thermionic vacuum arc procedure on Zircaloy -4 cladding. The surface coating characterization was performed using metallographic analysis and scanning electron microscopy (SEM) with an energy dispersive spectra detector (EDS), X-ray diffraction (XRD), and X-ray photoelectron spectroscopy (XPS) investigations. The thickness of the Cr coating determined from SEM images is around 500 nm layers After the autoclaving period, the thickness of the samples increased in time slowly. The kinetic of oxidation established a logarithmic oxidation law. The corrosion tests for various autoclaving periods of time include electrochemical impedance spectroscopy (EIS) and potentiodynamic tests, permitting computing porosity and efficiency of protection. All surface investigations sustain electrochemical results and promote the Cr coating on Zircaloy-4 alloy autoclaved for 3024 h as the best corrosion resistance based on decrease in corrosion current density values simultaneously with the increase of the time spent in autoclave. A slow increase of Vickers micro hardness was observed as a function of the autoclaved period as well. The value reached for 3024 h being 219 Kgf/mm$^2$ compared with 210 Kgf/mm$^2$ value before autoclaving.

**Keywords:** corrosion; Zircaloy-4; Cr coating; SEM; XRD; XPS; porosity; micro hardness





## 1. Introduction

The nuclear power generation industry has used Zr-based alloys, such as Zircaloy-2 (Zr-2) and Zircalloy-4 (Zy-4) alloys, as fuel cladding materials in light water reactors owing to their neutron transparency and corrosion resistance [1,2]. The 2011 Japan Earthquake and Tsunami in 2011, and the events that followed at the Fukushima Daiichi after this aggressive accident [3] led to a large interest in the world in development of fuels with enhanced performance during such rare events [4]. Starting 2012, accident tolerant fuel (ATF) development programs were developed in many research programs and industry teams [4,5]. Depending on the light water reactor (LWR) system details and accident development fuels with enhanced accident tolerance, zircaloy system are those that, compared with the currently used $UO_2$, can tolerate loss of active cooling in the reactor core for a much longer period of time [5]. Since welding is mandatorily applied to join end plugs to a cladding tube, various welding procedures with different susceptibilities to resistance during severe accidents are currently used.

However, the special attention of ATF development should be focused to estimate the oxidation behavior of Zr-based alloy welds, especially in the loss of coolant accident conditions.

In such context research investigation are focused on three main directions [6,7], including the following:

(a)　Enhancing of both chemical composition and technology fabrication processes for Zr alloys leading to advanced materials such as E635, ZIRLO, M5, MDA, HiFi, X5A, etc. [8].

(b)　Development of advanced alloys suitable for use in special event The ATF new candidate materials, for replacements of current Zr include refractory metals, advanced steels or SiCf/SiC [9].

(c)　development of new advanced coatings on existing alloys such as Zircaloy 4 which seems to be the most investigated direction, since no major change in design changes of LWR is required and is the most economically viable strategy [10–15].

In oxidizing environments, Zr is reported to form a several-nanometer-thick tetragonal oxide layer on its surface that protects the metal from further oxidation [6,7]. However, at elevated temperatures and pressures, this passive oxide layer continues to grow up to a critical thickness, beyond which the protective tetragonal phase transforms into a rapidly growing non-protective monoclinic oxide phase [16–18]. The unstable oxide growth causes internal stresses and crack formation, eventually resulting in coating breakdown and corrosion The main advantages of coated cladding [19] include:

-　Low neutron penalty and unmodified mechanical behavior if coatings are thinner than 20 μm;

-　Important decreases in corrosion kinetics for metallic and ceramic coatings;

-　a significant decrease and hydrogen embrittlement for such coatings due to reduced hydrogen pickup [17].

The appearance of water side corrosion of the fuel cladding and hydrogen uptake in the cladding at high burnup are the main limitations to burnup extension of nuclear fuel [19].

The selection of coating materials was based on the neutron cross-section, thermal conductivity, thermal expansion, melting point, phase transformation behavior, and high-temperature oxidation.

Oxidation resistant coatings are a relatively simple short-term solution to enhance the resistance of zirconium based nuclear fuel cladding to accelerated high temperature (HT) steam oxidation.

Various types of coatings have been proposed to improve the performance of current Zr-based alloy claddings, mainly regarding water side corrosion and high-temperature steam oxidation resistance, during normal and accident conditions. Cr-based coatings perform well under such conditions due to the successful growth of a protective $Cr_2O_3$. The major demerit observed initially was that $Cr_2O_3$ scale can only withstand the service temperature up to ~1200 °C when exposed to steam [19].

During these early studies, metallic chromium coatings have shown an encouraging behavior with excellent corrosion resistance in nominal conditions, and a significant enhancement of the HT steam oxidation behavior in accidental conditions, when compared with the reference uncoated zirconium based cladding materials [11]. More recently, the studies have confirmed the good behavior of Cr coatings in steam environment at HT, up to at least 1300 °C [19–22].

Other studies have also shown that Cr-coated zirconium alloy protects the underlying substrate from oxygen or hydrogen ingress and inhibits the formation of the brittle Zr-α(O) phase and leads to a significant improvement in the post-quench ductility of the cladding [23]. The ductility of the Cr-coated cladding is indeed similar before and after high temperature oxidation and therefore retains its integrity even after relatively long

exposure to high temperature steam for a 15 μm thick Cr coating up to at least 6000 s at 1200 °C [24].

Generally, it was concluded that regarding corrosion resistance and neutron stability, coatings such as Cr and CrN are the most promising. Both types of coatings are resistant to corrosion in LWR coolant and stable under neutron irradiation at expected temperatures, but Cr-coatings provide increased resistance to high temperature steam oxidation as well, while CrN does not [25]. CrN coatings demonstrate barrier properties at temperatures above 1350 °C, but only for thick interlayers. It is known that decomposition of CrN with formation of zirconium nitride, despite the barrier properties of ceramic coatings which are advantageous, are brittle, and small changes in temperature can lead to the formation of micro cracks and coating failures [20]. Musil proposed multilayers to prevent cracking CrN/Cr (250/250 nm) coatings against Cr-Zr inter-diffusion during heating up to 1400 °C [26,27].

The general coating techniques include physical vapor deposition (PVD), chemical vapor deposition (CVD), electrodeposition, thermal spray, cold spray, and pack cementation [20]. The techniques themselves can be modified and, in some cases, combined to produce the desired properties and behavior of the coatings.

The coating deposited using PVD is very dense and adherent and essential pore-free. In fact, despite that the high-temperature behavior of cladding materials such as Zr alloys coated with chromium was relatively largely investigated [28–31]. The microstructure, oxidation and corrosion behavior significantly vary based on the deposition techniques, deposition parameters, and thickness used.

Based on above recognized remarks, this paper presents the novelty of the testing corrosion behavior of Zy-4 coated with chromium by Thermionic Vacuum Arc (TVA) technique under CANDU reactor conditions.

In this system, a steady plasma plume is formed above the crucible containing the material to be deposited.

The specificity of this plasma originates in the kind of film precursors the source can provide on the substrate: only neutral atoms and highly energetic ions of the material are to be deposited, with no other particles (e.g., no buffer gas is used). Therefore, the resulting film is dense (due to the high energy ions; hundred eV), pure (no buffer gas is used) and very smooth (no droplets, due to the gentle mechanism of plasma ignition via electron evaporation). These film characteristics are relevant for corrosion resistance applications.

These particularities related to the kind and energy of precursor particles are not met simultaneously in other plasma sources. Therefore, they give great opportunities to deposit films having unique morphologies and composition. Other techniques similar to TVA, such as cathodic arc (CA) or pulsed laser deposition (PLD), either provide low energy ions (as in CA) or droplets are inherently formed (as in PLD).

It is to mention that even though microstructure and mechanical aspects of coated Zr alloys to improve their performances were widely investigated, their corrosion studies are scarcely and based especially on evolution of weight gain [32,33]. The recent relevant review devoted to protective coatings for accident tolerant Zr-based fuel claddings [20] introduced advanced latest results in the field of protective coatings for ATF cladding based on Zr alloys, presenting their behavior under normal and accident conditions in LWRs. The review attention has been focused on the protection and oxidation mechanisms of different coated cladding, including Cr coatings, as well as to the interdiffusion process between coatings and zirconium (these having less corrosion information available). The above comments and references sustained the need, the novelty, and relevance of the present research, which is an electrochemical approach comparing investigation of uncoated and coated corrosion behavior at normal operating conditions, lithiated water, 310 °C, pH = 10.5. The structure and properties of the developed vacuum-arc chromium coatings were investigated. It is shown that these coatings can be used as protective elements for existing fuel claddings, made of zirconium alloys, in light-water reactors of pressurized water reactor (PWR) and boiling water reactor (BWR) types.

## 2. Materials and Methods

### 2.1. Coating Material

A Zircaloy-4 tube alloy (see alloy composition in Table 1) was used as a substrate, as this material is used in fuel cladding tubes [34]. The Zircaloy-4 tube with an outer diameter of 13 mm and a wall thickness of 0.45 mm was cut into 20 mm sections and then halved lengthwise. Then, a 3.5 mm diameter hole was bored through one end of each half tube for the purpose of subsequently securing the sample during the autoclaving tests. Before the TVA, the substrates were ultrasonicated in isopropyl alcohol for 15 min, followed by nitrogen drying.

**Table 1.** Composition of zircaloy tube alloy (%).

| Alloying Elements, (wt.%) | | | | |
|---|---|---|---|---|
| Sn | Fe | Cr | O | Zr |
| 1.32 | 0.29 | 0.14 | 0.12 | Balance |

The selection of coating material was based on the neutron cross-section, thermal conductivity, thermal expansion, melting point, phase transformation behavior, and high-temperature oxidation rate. The metal base Cr was selected for coating the surface of Zy-4 alloy, as 1–3 mm Cr pellets of 99.99% purity procured from NEYCO company (Vanves, France).

### 2.2. Coating Method

Cr coatings studied in this paper have been deposited using TVA technique. The characteristics of this plasma source are presented with details in previous papers [35,36] and the equipment has been developed at the Laboratory of Low Temperature Plasma from National Institute for Laser, Plasma and Radiation Physics (NILPRP, Magurele, Romania).

The experimental arrangement of the TVA technique consists of a grounded cathode containing a Tungsten filament and an anode, a crucible containing the Cr pellets to be evaporated. A schematic view of the experimental setup is presented in Figure 1.

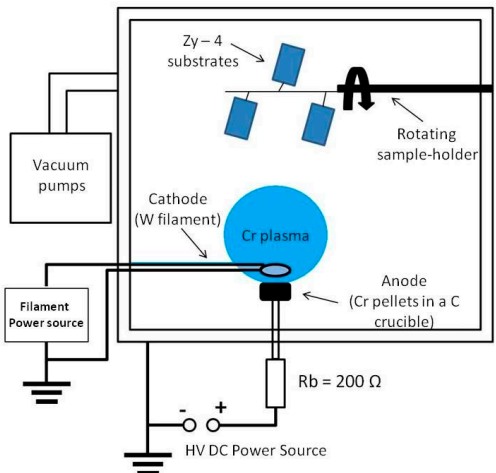

**Figure 1.** Schematic view of the experimental setup for TVA coating method used to apply a Cr layer on the Zy-4 surface.

The TVA plasma source developed in our NILPPR lab is a PVD technique capable to synthesize very compact and smooth thin films. The ignition of the TVA plasma is based on gentle evaporation of the material to be deposited using high energy electrons emitted by the cathode, followed by an avalanche of electron-atom collisions, under a vacuum of $10^{-6}$ torr or better. The TVA plasma is ignited at the moment when the chromium vapours

reach a sufficient pressure that gives a high probability of collision with electrons. Ions and an avalanche of electrons are thus formed above the crucible. The ions formed emerge from the plasma plume and travel in a straight direction towards the chamber walls due to the voltage difference between the plasma and the grounded chamber walls. The chromium vapors disperse from the plasma plume due to the pressure difference between the place where are created (above the anode) and the rest of the chamber which is vacuumed. Thus, the chromium film is obtained from atoms and ions of chromium only.

### 2.3. Coating Characterization

Zr clad is susceptible to corrosion from the coolant and fuel sides, due to the oxygen and hydrogen penetration. The oxidation and hydrogenation of zirconium in LWR are taking place according to reactions (1) and (2) as was evidenced in literature [37–41]. Coating of Zr with chromium introduced the third reaction (Equation (3)), representing Cr oxidation process which will complete. After oxidation, $Cr_2O_3$ exists on the surface of the Cr coating and $ZrO_2$ has been formed on the inner surface of Zr. At high temperatures, Zr and Cr reacted with water steam according to the Equations (1) and (2).

$$Zr_{(s)} + 2H_2O_{(g)} = ZrO_{2(s)} + 2H_{2(g)} \tag{1}$$

$$Zr + xH \rightarrow ZrH_x \tag{2}$$

$$Cr_{(s)} + 2H_2O_{(g)} = Cr_2O_{3(s)} + 3H_{2(g)} \tag{3}$$

After Cr deposition on Zy-4 alloy by the TVA method, the samples were exposed at 310 °C and 10 MPa to LiOH solution [42,43]. This electrolyte has a conductivity of 71 $\mu S \cdot cm^{-1}$ and a pH value of 10.5 at 25 °C. The dissolved oxygen content was maintained below 2 ppm by thermal degassing at 100 °C. Periodically, samples of Zy-4 covered with Cr were systematically removed from the autoclave for weight gain measurements and morphological and structural analysis as well as electrochemical testing. The autoclaved solution was replaced with a fresh solution after each inspection. After cleaning with acetone and drying, the weight gain of each sample was measured using a balance, providing a precision of $\pm 10^{-4}$ mg.

### 2.3.1. Morphological and Structural Surface Analysis

First, the coated and uncoated Zy-4 sample surface were characterized using metallographic analysis, SEM and XPS. To highlight the hydrides, metallographic analysis was performed using the Olympus GX 71 optical microscope. The hydrides have been highlighted by chemical etching in a solution composed of 45 mL $HNO_3$ (67%), 45 mL $H_2O_2$ (30%) and 7 mL HF (30%) for 30 s. The Vickers microhardness ($MHV_{0.1}$) was determined by an OPL tester in an automatic cycle.

For the morphological characterization scanning electron microscopy (SEM, Hitachi SU 8230 scanning electron microscope) at a pressure of 0.7 torrs at 10 and/or 15 kV were used. To identify the elemental composition of the samples, an energy dispersive spectra detector (EDS) was used.

The XRD patterns of the investigated samples were obtained using a D8 ADVANCE (Brucker-ASX) X-ray Diffractometer in a θ–2θ geometry using CuKα radiation ($\lambda$ = 1.5406 Å) and operating at room temperature. X-ray diffraction measurements of all samples were carried out in the 5°–140° range. The identification of the phase was made by referring to the International Center for Diffraction Data—ICDD (PDF-2) database.

XPS measurements were performed on autoclaved chromium coated Zy-4 alloys using an ESCALAB XI + X-ray Photoelectron Spectrometer (XPS) Microprobe. Monochromatic Al Kα radiation (1486.6 eV) and 900 μm X-ray spot size was used. The acquired spectra were calibrated with respect to the C1s line of surface adventitious carbon at 284.8 eV. An electron flood gun was used to compensate for the charging effect in insulating samples.

### 2.3.2. Electrochemical Tests

An electrochemical measurement system PARSTAT 2273 (Princeton Applied Research, AMETEK, OakRidge, TN, USA) was used to perform all of the electrochemical measurements. Also, a classical three-electrode electrochemical cell consisting of a working electrode (Cr-Zy-4 sample), a saturated calomel reference electrode (SCE) and two auxiliary electrodes (graphite rods) was used to perform electrochemical measurements. All electrochemical tests were performed in triplicate at room temperature ($22 \pm 2$ °C). The electrochemical parameters were obtained by the open circuit potential variation, electrochemical impedance spectroscopy and potentiodynamic measurements.

The susceptibility of the material to generalized corrosion was observed by open circuit potential measurements in a specific primary circuit solution (LiOH solution, pH 10.5).

Electrochemical impedance spectroscopy (EIS) tests were performed in a chemically inert solution with pH = 7.26 (0.05 M boric acid with 0.001 M borax solution), at open circuit potential (OCP) with an amplitude of 10 mV in the frequency range from 100 mHz to 100 kHz after OCP stabilization. The experimental EIS results were simulated with equivalent electrical circuits as appropriate models using ZView 2.90c software (Scribner Associates Inc., Southern Pines, NC, USA).

Potentiodynamic measurements were made in LiOH solution at pH 10.5 with a scan rate of 0.5 mV·s$^{-1}$ and a range from $-250$ to 1000 mV relative to the open current potential (OCP). In CANDU primary circuit LiOH is added to the cooling water to adjust an alkaline pH. Due to its technological relevance the influence of LiOH on the corrosion of zirconium and its alloys has been investigated by other authors [44].

## 3. Results and Discussion

### 3.1. Oxidation Kinetics

For the chromium coated Zy-4 samples were performed weight gain measurements after different autoclaving period carried out at 310 °C and 10 MPa. Figure 2 presents the weight gain data as a function of exposure time for Cr coated Zy-4 samples in LiOH solution.

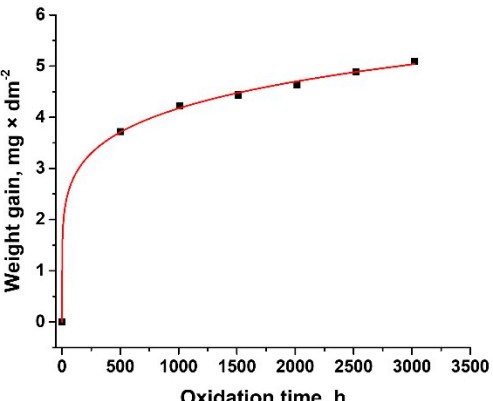

**Figure 2.** Oxidation kinetics of chromium coated Zy-4 samples in the LiOH solution at 310 °C and 10 MPa.

As can be seen from Figure 2, all samples gained weight during oxidation.

A typical power rate equation was used to study the oxidation process, described by:

$$\Delta W = k_p \times t^n \tag{4}$$

where $\Delta W$ is the oxide weight gain (mg/dm$^2$), k$_p$ is the rate constant, t is exposure time (h) and $n$ is the exponent. Fitting the data with Equation (4) the oxidation constants were determined. In Table 2 are presented the obtained $k_p$ value and R-squared value of trend

($R^2$) of the plot. The value of the correlation coefficient ($R^2$ = 0.999) indicates that the experimental Cr oxidation results fit well.

**Table 2.** Kinetic parameters for Cr coated Zy-4 samples.

| Kinetic Equation | $k_p$ | $n$ | $R^2$ |
|---|---|---|---|
| $y = 1.286 \times t^{0.17}$ | 1.286 | 0.17 | 0.999 |

Some researchers obtained a value for $n$ of 0.5 which indicates a parabolic law [28], whereas others have observed quartic oxidation kinetics (exponent $n \approx 0.25$), we got a value of 0.17 for $n$. This value indicates that the oxidation kinetics of Cr coating on Zy-4 samples does not follow a parabolic or quartic law perfectly. According to Ma et al. [19], in our case de oxidation kinetics for chromium coating in zircaloy follow logarithmic kinetics.

Based on the values obtained, we can calculate the thickness of the oxide scale from the weight gain of the sample. Based on the values obtained from the weighing of the samples, the oxide thickness was calculated dation. The oxide film thickness at different periods of oxidation for Cr coated Zy-4 samples was calculated from the weight gain according to Equation (5) [12].

$$d \times \rho = \Delta W \times \frac{x \times M_{Cr} + y \times M_{O_2}}{y \times M_{O_2}} \qquad (5)$$

where $d$ is oxide film thickness, $\rho$ is the theoretical density of oxide, $\Delta W$ is weight gain per unit area, $x$ and $y$ are the coefficients in the metal oxide $M_xO_y$, $M_{Cr}$ and $M_{O2}$ are referred to the relative molecular mass of metal element and oxygen, respectively.

Figure 3 shows the oxide thickness behavior of the Cr coated Zy-4 samples in the LiOH solution at 310 °C and 10 MPa.

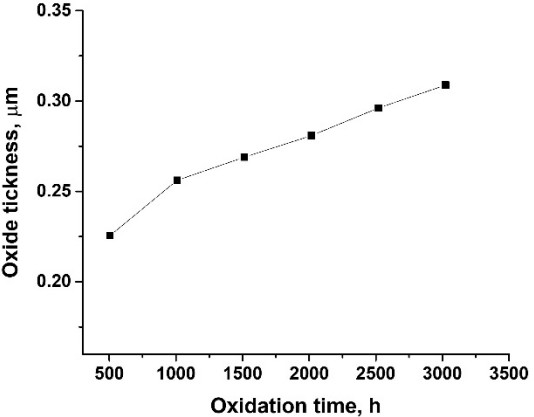

**Figure 3.** Oxide thickness variation on exposure time for Cr coated Zy-4 samples in a LiOH solution.

From Figure 3, we can see that the oxide thickness of all studied samples increased with the oxidation time, reaching after 3024 h of autoclaving, an average thickness of oxide film of approximately 0.3 μm much lower compared to the values obtained under the same conditions for the uncovered zircaloy samples [37].

### 3.2. Morphological and Structural Characterization

### 3.2.1. Metallographic Analysis (Optical Microscopy)

The microstructural characterization indicated that hydrides with a relatively uniform distribution were precipitated on the horizontal orientation across the alloy in concordance with hydride precipitation discussed in literature [41,45–47]. In general, the fraction of absorbed H increases with exposure time. It was shown that there was a significant increase in H solubility with temperature from 0.01 at.% at room temperature, to 1.5 at.% at 300 °C and around 5.6 at.% at 500 °C [38,39]. It is known that there is a presence of four phases of

Zr hydrides with different crystal structures, but many experimental studies done to date have been on δ and γ phases, since they are the phases most liable for embrittlement and fracture of materials. The fraction of δ and γ hydrides is related to the H concentration and cooling rate of Zr. A higher concentration of H and/or a slower cooling rate will lead to more δ hydrides, while the opposite will lead to more γ hydrides [40]. The relatively slow cooling rates during the normal operation of nuclear fuel rod claddings lead to δ hydrides formation, which causes most of DHC (Delayed Hydride Cracking) under actual PHWR's operating conditions.

Figure 4 shows the density evolution of hydrides with autoclaving time, and we can observe that there is not an obviously grown of the density of hydrides from 504 testing hours to 1512. The inhibition of hydride precipitation can be explained by the Cr coating layer deposition. For 3024 testing hours it can be seen a slowly increment of the density of hydrides, which can be explained by the grow of new layer of grains until the stress accumulation has led to the formation of pores and cracks that break the protective properties of the layer to the corrosive species.

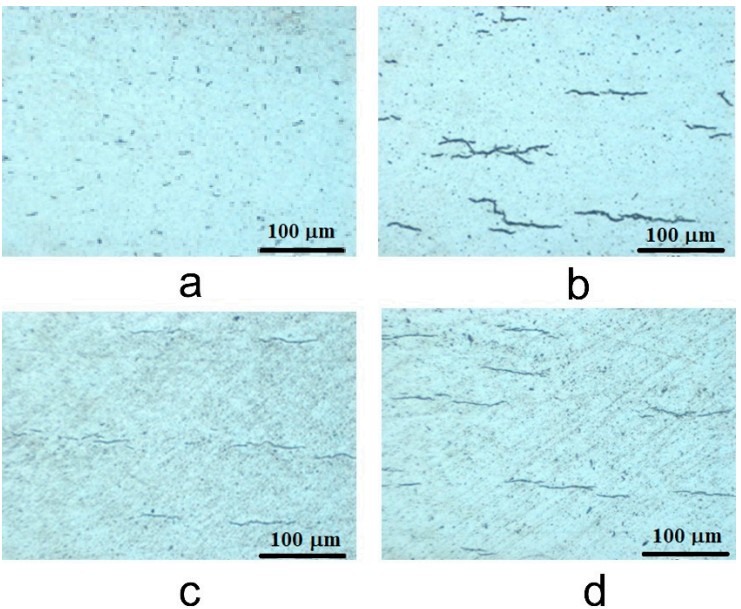

**Figure 4.** Representative Zy-4 hydride formation for different autoclaving period: (**a**) 0 h; (**b**) 504 h; (**c**) 1512 h; (**d**) 3024 h.

The Vickers microhardness was determine using an OPL tester in automatic cycle at a load of 0.1 Kg and an indentation dwelling time of 10 s. An average of 10 indents was taken for each sample. The microhardness of the Cr coated Zy-4 was 210 Kgf/mm$^2$, which slowly increased to 212 Kgf/mm$^2$ for the Cr coated Zy-4 autoclaved for 504 h. For the Cr coated Zy-4 autoclaved for 1512 h, the microhardness was further elevated to 214 Kgf/mm$^2$, and finally to 219 Kgf/mm$^2$ for 3024 h of autoclave testing. This slowly increase in microhardness values is ascribed to the matrix hardening effect induced by hydrides that are embedded in the Zy-4 matrix.

3.2.2. Scanning Electron Microscopy (SEM) Measurements

More information on the morphology of chromium coated Zy-4 samples before and after autoclaving was obtained using SEM microscopy. Five measurements were performed each time. Figure 5 presents the SEM on cross-sections of the samples before (Figure 5a) and after (Figure 5b) autoclaving, the thickness of the coating layer present on the surface of each sample could be measured. As presented in SEM images from Figure 5a,b before and after autoclaving period the thickness values are around 500 nm and respectively 800 nm. The presence of chromium or oxide layer on the surface of each sample being

studied can be observed as an inset of each imagine from Figure 5. These show the EDS compositional profiles of cross-sections through the coating grown on the Zy-4 samples.

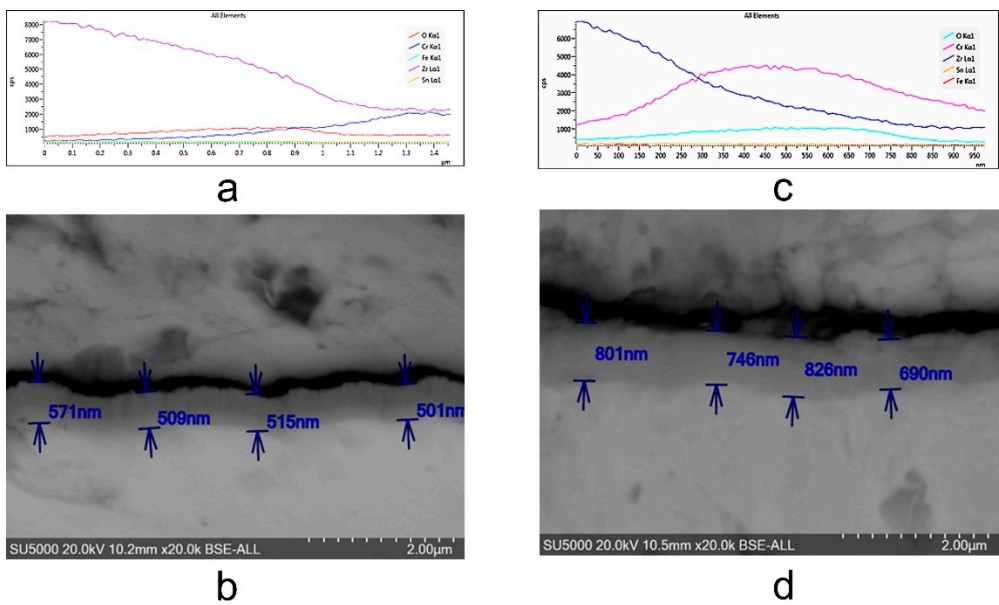

**Figure 5.** The SEM cross-section for chromium coated Zy-4 samples before (**b**) and after (**d**) autoclaving period; (**a**,**c**) EDS profile of a cross-section through the coating layer grown on an Zy-4 sample.

### 3.2.3. XRD Measurements

The XRD diffractogram presented in Figure 6a for both uncoated Zy-4 substrate (blue line) and TVA Cr plasma coated Zy-4 substrate (red line) revealed the appearance of a crystalline phase of Chromium with a 110 preferred orientation according to the PDF—4+ file No. 04-008-5987.

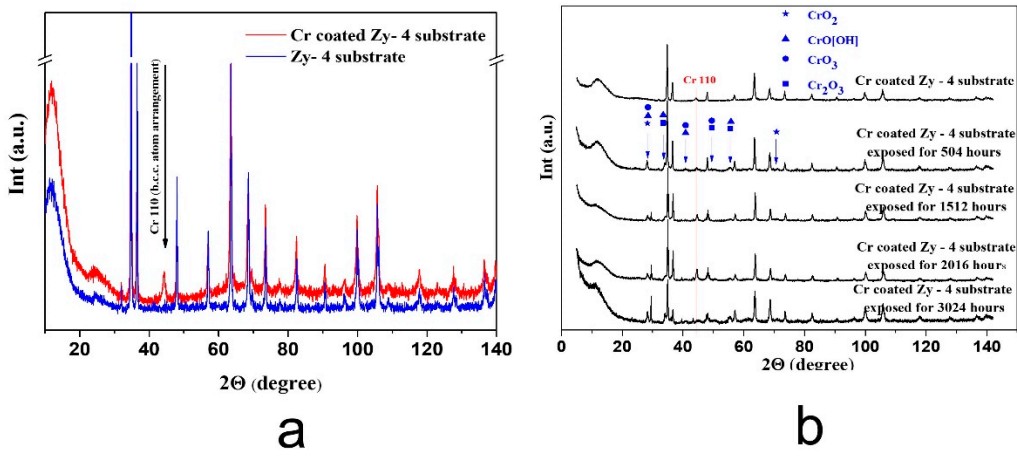

**Figure 6.** (**a**) XRD patterns of uncoated (blue line) and chromium coated (red line) Zy-4 substrate evidencing the appearance of Cr 110 characteristic peak; (**b**) evolution XRD patterns of the chromium coated Zy-4 samples as a function of exposure time.

Fitting the XRD patterns of both Zy-4 substrate and Chromium coated Zy-4 substrate using the TOPAS XRD software the crystallites size was determined as 100 nm for the Zy-4 alloy and 15 nm for the Chromium. The XRD patterns from Figure 6b present the modifications induced by different autoclaving period. It can be observed that increasing autoclaving time patterns characteristic to both chromium oxides such as $CrO_2$ (PDF—4+ file No. 04-007-5823), $CrO_3$ (PDF—4+ file No 04-007-2455) and $Cr_2O_3$ (PDF—4+ file No. 04-

007-4822) and CrO[OH] hydroxide (PDF—4+ file No. 04-010-0689) appear in the collected spectra at 2θ = 28°, 34°, 41°, 50°, 55°, 71° marked by blue arrows. Moreover, their peaks intensities increase with autoclaving time, suggesting a thickness increase.

### 3.2.4. XPS Measurements

Figure 7a presents XPS survey spectra of the following samples: zircalloy substrate, Cr film deposited on Zy-4 alloy, and of autoclaved chromium coated Zy-4 alloy for the specified time intervals (in days). As could be observed in the spectra, Cr is present in all samples.

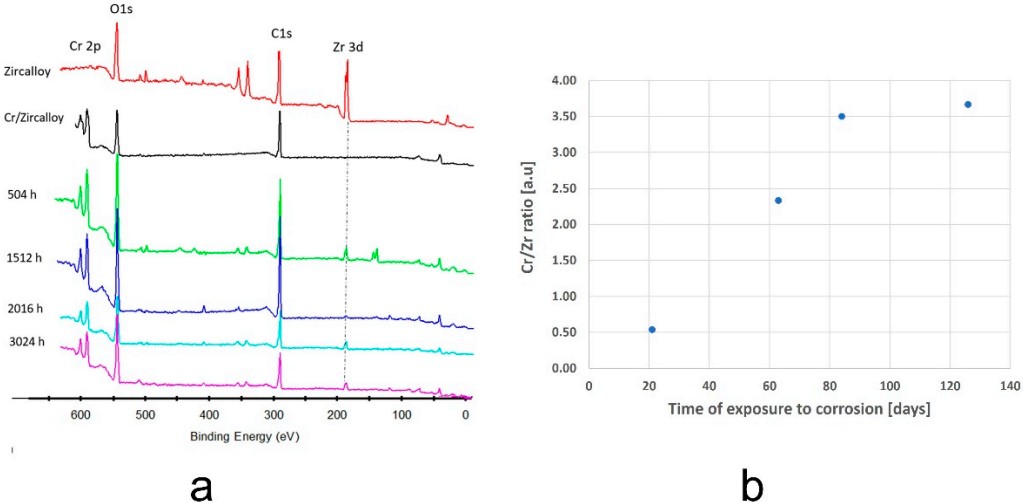

a　　　　　　　　　　　　　　　　　　　b

**Figure 7.** (**a**) XPS survey spectra of Zy-4 substrate, Cr film deposited on Zy-4 alloy and of autoclaved Cr/Zy-4 alloy for the specified time intervals (in days); (**b**) variation of Cr/Zr ratio with autoclaving time.

By quantification of XPS data, signal variation of Cr to Zr is also presented (Figure 7b). The evolution of Cr/Zr a.u. ratio in time sustain the type of increase of oxide thickness on exposure time for Cr coated Zy-4 samples presented in Figure 3.

Figure 8 gives XPS spectra of Cr1*s* window acquired for deposited Zircalloy and for autoclaved samples. As can be observed in this figure, the XPS signal from Cr metal completely disappeared on all autoclaved samples and only Cr oxides are present.

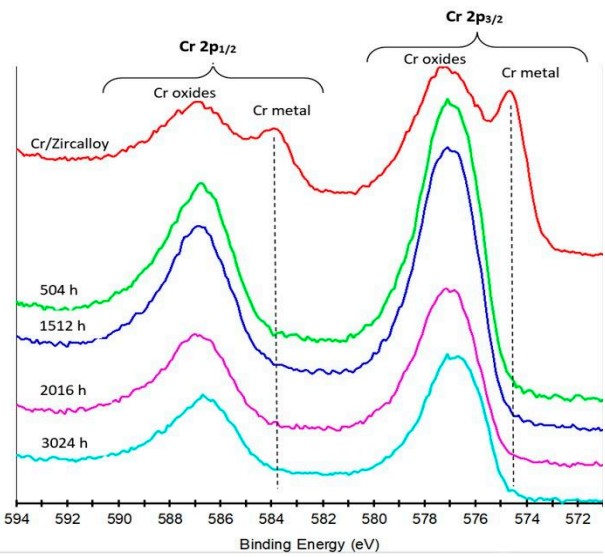

**Figure 8.** XPS spectra of Cr1*s* window for coated samples autoclaved for the specified time intervals.

### 3.3. Electrochemical Characterization

3.3.1. Open Circuit Potential Measurements

In order to predict the generalized corrosion behavior of Cr coated Zy-4 samples, the open circuit potential (OCP) variation was recorded. In Figure 9, we see that that for all studied samples, the value of the open circuit potential is relatively stable.

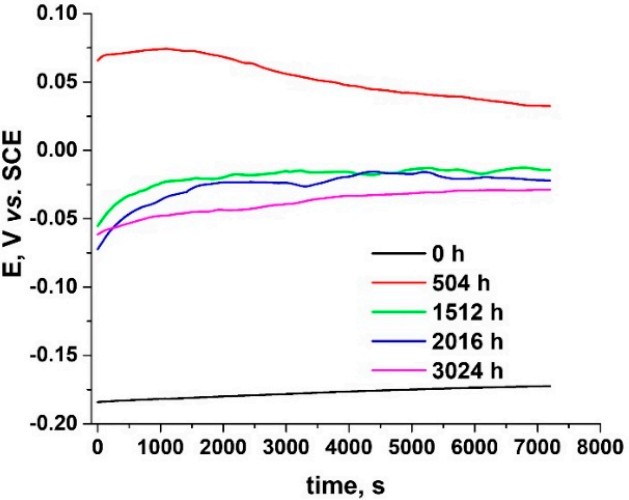

**Figure 9.** Open circuit potential variation for Cr coated Zy-4 alloy before and after autoclaving for 504, 1512 or 3024 h in LiOH solution at 310 °C and 10 MPa.

The most electronegative value for OCP ($-172$ mV) were recorded for the sample covered with Cr but not autoclaved. For the zircalloy coated samples, after autoclaving, the stabilized OCP values were obtained at 32 mV for the sample oxidized for 504 h, and around $-20$ mV for autoclaved samples for 1512, 2016, and 3024 h, respectively, with a slight shift towards more electronegative values as the autoclaving time increases. In all cases studied, the values obtained for OCP are more electropositive than those recorded for the uncovered alloy but subjected to autoclaving under the same conditions [37]. Generally, the change in the open circuit potential can indicate the corrosion state and corrosion behavior of the material surface and it is expected, that the open circuit potential moves in the positive direction and the corrosion rate decrease gradually [48]. The difference between the OCP of non-autoclaved sample and the sample autoclaved for 504 h we can consider to be the result of a chromium oxide formation during autoclaving, which is adherent and may cover eventual pores or discontinuities in the chromium layer. In time with the thickness evolution of chromium layer, the differences are less significant and the differences in the OCP seems to be less significant as well.

In agreement with these values, we can say that the coating improves the corrosion behavior of Zy-4 samples.

3.3.2. Electrochemical Impedance Spectroscopy

The second electrochemical method used to analyze the protective properties of Cr coating on the Zy-4 alloy was electrochemical impedance spectroscopy [49]. The Nyquist and Bode diagrams from Figure 10 show the spectra recorded at open circuit potential after 10 min of immersion in LiOH solution for the Cr-coated Zy-4 alloy by TVA method.

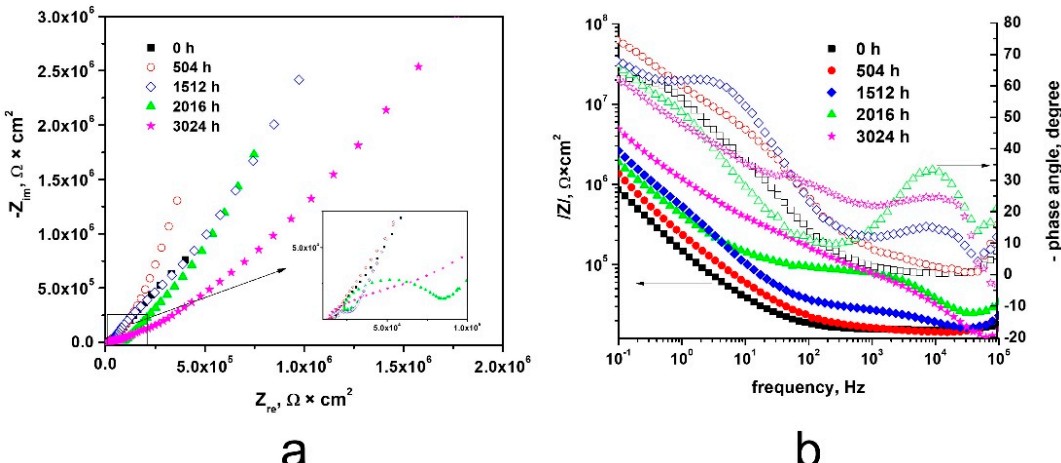

**Figure 10.** Nyquist (**a**) and Bode (**b**) diagrams for Cr coated Zy-4 alloy after different autoclaving period in LiOH solution at 310 °C and 10 MPa.

As we can see from the Nyquist diagrams (Figure 10a), a single open capacitive appeared for non-autoclaved sample, while for all autoclaved Cr coated Zy-4 samples two capacitive semicircles appear corresponding to the two interfaces created. For all of the autoclaved samples, higher values of the capacitive semicircle diameter were recorded compared to the non-autoclaved Cr coated zircalloy.

From the Bode plots, Figure 10b, we can observe that for the coated Zy-4 samples subjected to oxidation higher impedance values were obtained. The impedance, |Z|, values being directly proportional to oxide resistance, we can say that these high values obtained for impedance modulus indicate a good corrosion resistance of these samples. From the Bode diagram (Figure 10b) we can see that two maxima of the phase angle are obtained, one at high frequencies and the other at low frequencies.

All experimental EIS data were fitted using the equivalent electrical circuit model [50] from Figure 11.

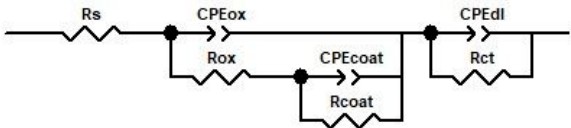

**Figure 11.** Equivalent circuit diagram for the Chromium coated Zy-4 samples after different autoclaved period.

In this equivalent electrical circuit, Rs is the solution resistance between the electrode and the electrolyte; CPEox is the constant phase element; Rox the resistance of the oxide layer; CPEcoat the constant phase element; Rcoat the resistance of the coating layer; CPEdl the constant phase element for double layer; and Rct charge transfer resistance. We decided to use constant phase elements (CPE) instead to explain the deviation of the capacitances between the actual measurements and the ideal pure capacitances, due to the local inhomogeneities of the dielectric material, surface roughness, and relaxation effect; the degree of deviation from the ideal capacitance depends on the value of $n$ ($0 < n < 1$). As can be seen from Table 3, in which are presented all the parameters resulting from the fitting of the experimental data with the proposed equivalent electrical circuit, a good fit of the data to this model was obtained.

**Table 3.** The values of equivalent electrical circuits elements for Cr coated Zy-4 alloy unoxidized and oxidized, after different periods in our LiOH solution at 310 °C and 10 MPa.

| Oxidation Period, h | $R_s$, $\Omega \times cm^2$ | $CPE_{ox}$-T $nF \times cm^{-2}$ | $CPE_{ox}$-P | $R_{ox}$ $K\Omega \times cm^2$ | $CPE_{coat}$-T $\mu F \times cm^{-2}$ | $CPE_{coat}$-P | $R_{coat}$ $\Omega \times cm^2$ | $CPE_{dl}$-T $\mu F \times cm^{-2}$ | $CPE_{dl}$-P | $R_{ct}$ $\Omega \times cm^2$ | Chi-Squared |
|---|---|---|---|---|---|---|---|---|---|---|---|
| 0 | 158.5 | – | – | – | 1.51 | 0.89 | $8.62 \times 10^5$ | 1.97 | 0.76 | 26,974 | $2.7 \times 10^{-3}$ |
| 504 | 158.7 | 0.161 | 0.98 | 1.25 | 1.93 | 0.63 | $1.97 \times 10^5$ | 1.81 | 0.94 | $2.89 \times 10^7$ | $4.1 \times 10^{-3}$ |
| 1512 | 143.2 | 1.35 | 0.93 | 57.97 | 1.21 | 0.79 | $1.81 \times 10^6$ | 0.122 | 0.98 | $3.82 \times 10^{10}$ | $1.1 \times 10^{-3}$ |
| 2016 | 170.7 | 9.19 | 0.9 | 11.78 | 0.89 | 0.73 | $1.13 \times 10^6$ | 0.766 | 0.99 | $1.94 \times 10^7$ | $8.9 \times 10^{-4}$ |
| 3024 | 152.3 | 3.56 | 0.88 | 46.06 | 0.94 | 0.78 | 56,051 | 0.295 | 0.69 | $4.9 \times 10^{14}$ | $1.8 \times 10^{-3}$ |

### 3.3.3. Potentiodynamic Polarization Tests

In order to see the Cr coated Zy-4 alloy behavior in a specific primary circuit solution (LiOH solution, pH 10.5), the potentiodynamic polarization curves of the sample before and after different autoclaving periods were recorded and are illustrated in Figure 12.

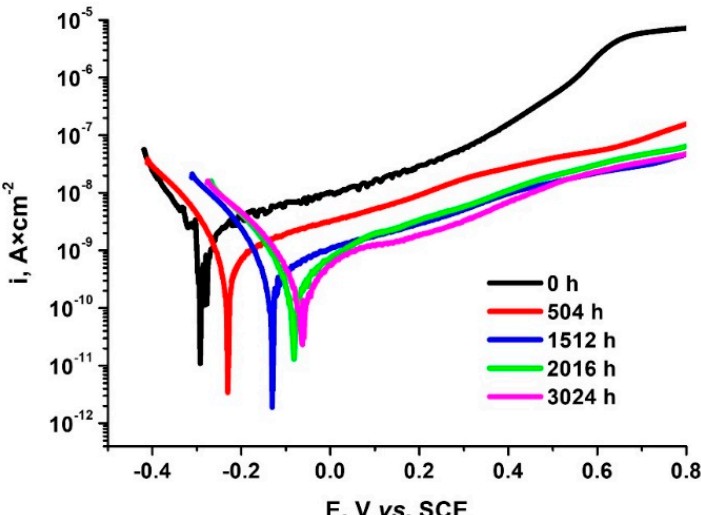

**Figure 12.** Polarization curves for Cr coated Zy-4 alloy before and after autoclaving for different period in LiOH solution (310 °C and 10 MPa) at a scan rate of 0.5 mV·s$^{-1}$.

Applying the Tafel slope extrapolation method for the polarization curves from Figure 12, the corrosion kinetic parameters shown in Table 4 were obtained. The main parameters are the corrosion potential ($E_{corr}$), corrosion rate ($V_{corr}$), current density ($i_{corr}$), polarization resistance ($Rp$), protection efficiency ($P_i$) and oxide porosity ($P$). Statistical analyses were performed for all parameters, and they are presented as mean $\pm 1$ standard deviation.

**Table 4.** Polarization parameters of Cr coated Zy-4 alloy autoclaved for different periods in LiOH solution (310 °C, 10 MPa).

| Sample after Different Autoclaving Time, h | $E_{corr}$, mV | $i_{corr}$, nA $\times$ cm$^{-2}$ | $V_{corr}$ nm $\times$ Year$^{-1}$ | $R_p$ M$\Omega \times$ cm$^2$ | $P_i$ (%) | $P$ (%) |
|---|---|---|---|---|---|---|
| 0 | $-283 \pm 0.02$ | $1.19 \pm 0.02$ | $1.43 \pm 0.02$ | $5.7 \pm 0.03$ | – | – |
| 504 | $-225 \pm 0.02$ | $0.492 \pm 0.01$ | $5.93 \pm 0.02$ | $7.2 \pm 0.07$ | $58.65 \pm 0.01$ | $0.0124 \pm 0.01$ |
| 1512 | $-119 \pm 0.01$ | $0.147 \pm 0.01$ | $1.77 \pm 0.02$ | $24 \pm 0.05$ | $87.65 \pm 0.01$ | $1.25 \times 10^{-4} \pm 0.01$ |
| 2016 | $-74 \pm 0.01$ | $0.107 \pm 0.01$ | $1.29 \pm 0.02$ | $33 \pm 0.07$ | $91.01 \pm 0.01$ | $2.16 \times 10^{-5} \pm 0.01$ |
| 3024 | $-59 \pm 0.01$ | $0.0981 \pm 0.01$ | $1.1 \pm 0.02$ | $39 \pm 0.07$ | $91.76 \pm 0.01$ | $1.13 \times 10^{-5} \pm 0.01$ |

The corrosion rate was calculated using Equation (6).

$$V_{corr} = \frac{K \times i_{corr} \times \text{EW}}{\rho} \tag{6}$$

In this Equation, $V_{corr}$ is corrosion rate (mm/year), $K$ is a constant, his value is $3.27 \times 10^{-3}$, mm $\times$ g/($\mu$A $\times$ cm $\times$ year), $i_{corr}$ is corrosion current density in $\mu$A/cm$^2$, EW is Equivalent Weight of alloy and $\rho$ is the density of Zircaloy-4 at room temperature [51,52]. The protection efficiency ($P_i$) has been evaluated quantitatively using Equation (7):

$$P_i = \left(1 - \frac{i_{corr}}{i_{corr}^0}\right) \times 100 \tag{7}$$

If the oxide grown on the Cr-coated zircaloy is electrochemically inert at low anodic overpotential, the porosity of the oxide can be estimated using the Equation (8):

$$P = \frac{R_P}{R_{P_{coat}}} \times 10^{-\left(\frac{\Delta E_{corr}}{\beta a}\right)} \qquad (8)$$

where $P$ is the total oxide porosity, $Rp$ is the polarization resistance of the substrate, $Rp_{coat}$ is the polarization resistance of the coated zircalloy, $\Delta E_{corr}$ is the difference potential between the free corrosion potentials of the coated zircalloy and the substrate, and $\beta_a$ is the anodic Tafel slope for the substrate. Using Equation (7), oxide porosity ($P$) has been evaluated quantitatively and the obtained results are presented in Table 4 [53].

As can be seen from Figure 12 and Table 4, for all autoclaved Cr coated Zy-4 samples, lower values of corrosion current density were obtained compared to the coated sample.

It can be observed that, with the increase of the autoclaving time, the values of the corrosion current density decrease slightly and, consequently, the corrosion rate decreases.

It can also be seen that as the autoclaving time increases, the corrosion potential shifts to more electropositive values. In the corrosion of coated materials, an important parameter is the protection efficiency ($Pi$) since it allows us to appreciate the integrity of the coating. In the case of the studied samples, the protection efficiency is high, this agreeing with the other obtained results. Comparing the results obtained with those obtained for uncoated Zy-4 samples conditions [37], in all the studied cases, the corrosion rate values are lower, indicating that the coating of the Cr alloy is beneficial in terms of corrosion resistance in the studied electrolyte.

According to the corrosion classification and definition [54] the main degradation mechanism identified being a generalized oxidation of the coating on the whole surface is a general corrosion attack. The presence of chromium oxide has been identified by various applied techniques. We didn't find local sites of oxidation, or any signs of spallation. The thickness evolution in time and micro hardness as well are small and do not sustain appearance of significant local corrosion types such as pitting. It can also be seen that the best protection against corrosion was provided by the sample that was autoclaved for 3024 h. The lowest porosity value for the oxide layer was obtained for the autoclaved sample during 3024 h which reconfirms the lowest value of the corrosion current density obtained for this sample.

## 4. Conclusions

Based on the experimental data, we can present the following performances achieved after autoclaved for various periods of time of TVA Cr coatings of Zircaloy 4:

a.  The thickness of samples determined from SEM images before autoclaving is around 500 nm; after autoclaving the period layers increase, the kinetic of oxides growth being a logarithmic one compared to the kinetics of the uncoated sample which is a parabolic one. The evolution of ratio Cr/Zr a.u. in time based on XPS experimental data sustain this type of increase.

b.  The XRD determinations indicated the appearance of a crystalline phase of Chromium with a 110 preferred orientation. In the case of autoclaved samples, patterns characteristic to chromium oxides and hydroxides appear in the collected spectra can be observed.

c.  Decrease in corrosion current density values simultaneously with the increase of the time spent in autoclave reaching the best value for 3024 h A shift to more positive values of corrosion potential and was identified at the same time. For 3024 h autoclaving the protection coating efficiency reached the highest value, being 91.76%

d.  High corrosion resistance demonstrated by Tafel plots is supported with highest impedance obtained in EIS experiments for 3024 h. Comparing the Nyquist diagrams for non-autoclaved sample, a single open capacitive appeared, while for all autoclaved Cr coated Zy-4 samples two capacitive semicircles are present, sustaining the interfaces created. Also, for all autoclaved samples, higher values of the capaci-

tive semicircle diameter were recorded compared to the non-autoclaved Cr coated zircalloy.

e. All surface investigations sustain electrochemical results and promote the Cr coating on Zircaloy-4 alloy autoclaved for 3024 h as the one with best corrosion resistance based on decrease in corrosion current density values simultaneously with the increase of the time spent in autoclave.

As future research, it will be useful in to get more knowledge about coatings evolution investigating corrosion aspects for various coatings thickness and for longer periods of time.

**Author Contributions:** Conceptualization, M.F., C.C.S.-B., and I.D.; methodology, D.D., F.G., and A.A.; software, D.D.; validation, F.G., M.F., and I.D.; formal analysis, D.D. and A.A.; investigation, F.G. and C.C.S.-B.; resources, A.A.; data curation, D.D. and A.A.; writing—original draft preparation, D.D., C.C.S.-B., and A.A.; writing—review and editing, F.G., M.F., and I.D.; visualization, C.C.S.-B., M.F., and F.G.; supervision, I.D.; project administration, A.A.; funding acquisition, F.G. All authors have read and agreed to the published version of the manuscript.

**Funding:** This research was funded by Romanian Ministry of National Education, UEFISCDI, Grant. No. NECOMAT-326PED/2020.

**Institutional Review Board Statement:** Not applicable.

**Informed Consent Statement:** Not applicable.

**Data Availability Statement:** The data presented in this study are available upon request.

**Conflicts of Interest:** The authors declare no conflict of interest.

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
