# Peer review of "Corrosion Behavior of Chromium Coated Zy-4 Cladding under CANDU Primary Circuit Conditions"

_coatings, doi:10.3390/coatings11111417_

Round 1

Reviewer 1 Report

The article is devoted to the study of the mechanisms of corrosion of chromium-based coatings deposited on zircaloy. This line of research is one of the promising areas of development of nuclear materials used to protect against the negative effects of ionizing radiation. However, despite all the positive aspects of the work, before accepting it for publication, the authors should answer a number of questions that arose when reading this work.
1. In the introduction, the authors should more accurately prescribe the relevance of this work in comparison with previously done work in this direction.
2. According to X-ray diffraction, the presence of oxide and hydroxide compounds is observed on the diffractograms, the authors need to indicate the phase relationship of these phases, and the dynamics of their change.
3. What is the basis for the choice of LiOH solution for carrying out corrosion testing experiments?
4. Authors should describe in more detail the mechanisms and type of corrosion as a result of external influences, depending on the time of testing.
5. Technical notes include the following. Authors should indicate measurement errors on all graphs.

Reviewer 2 Report

This manuscript is a thorough investigation of the corrosion behaviour of chromium coated Zircaloy 4. The topic of this paper is interesting from both an industrial and scientific point of view. The paper is relatively well-drafted and scientifically sound.

My main comments and observations on this manuscripts as follows:

In figure 5, the inset of EDS profile can be hardly seen. Please make it more legible.

For the evaluation of XRD measurements, the corresponding JCPDS files of all the detected phases should be given.

It would increase the quality of paper, if the XPS results are evaluated in a more detailed way.

For the open circuit measurements, the huge different between the OCP of non-autoclaved sample and the sample autoclaved for 504 hours is not explained, there is no consistency in the changes.

As for the Nyquist diagrams, as I can see, the diameter of capacitive curve for 504 h autoclaved sample is higher, than in the case of non-autocalved sample. In the text it is discussed otherwise.

Why the Authors chose the given type of EC for fitting the EIS data? Any reference for similar measurements?

Why the LiOH solution resistance is so high? In my opinion, the 15 kOhm solution resistance is not credible. Moreover, as I have read one of your most recent article "Diniasi, D.; Golgovici, F.; Marin, A.H.; Negrea, A.D.; Fulger, M.; Demetrescu, I. Long-Term Corrosion Testing of Zy-4 in a LiOH Solution
under High Pressure and Temperature Conditions. Materials 2021, 14, 4586." the topic is very similar, and in this case only two time constant EC were used.

For the potentiodynamic measurements, the calculation method, formula for Vcorr is not given.

Thorough language check is also needed.

Round 2

Reviewer 1 Report

The authors made corrections to the article in accordance with all the comments of the reviewer.

Reviewer 2 Report

The manuscript have improved a lot after the revision. The answers given to the questions and comments are correct and satisfactory.

In this state now it is worthy of publishing.